# Examining Clinical Opinion and Experience Regarding Utilization of Plain Radiography of the Spine: Evidence from Surveying the Chiropractic Profession

**DOI:** 10.3390/jcm12062169

**Published:** 2023-03-10

**Authors:** Philip A. Arnone, Steven J. Kraus, Derek Farmen, Douglas F. Lightstone, Jason Jaeger, Christine Theodossis

**Affiliations:** 1The Balanced Body Center, Matthews, NC 28105, USA; 2Biokinemetrics, Inc., Carroll, IA 51401, USA; 3Institute for Spinal Health and Performance, Folsom, CA 95630, USA; 4Community Based Internship Program, Associate Faculty, Southern California University of Health Sciences, Whittier, CA 90604, USA; 5Chair, Radiology Department, Sherman College of Chiropractic, Boiling Springs, SC 29316, USA

**Keywords:** X-ray utilization, Evidence-Based Practice (EBP), chiropractic practice, clinical opinion, chiropractic survey, radiographs

## Abstract

Plain Radiography of the spine (PROTS) is utilized in many forms of healthcare including the chiropractic profession; however, the literature reflects conflicting opinions regarding utilization and value. Despite being an essential part of Evidence-Based Practice (EBP), few studies assess Doctors of Chiropractic (DCs) clinical opinions and experience regarding the utilization of (PROTS) in practice. In this study, DCs were surveyed regarding utilization of PROTS in practice. The survey was administered to an estimated 50,000 licensed DCs by email. A total of 4301 surveys were completed, of which 3641 were United States (US) DCs. The Clinician Opinion and Experience on Chiropractic Radiography (COECR) scale was designed to analyze survey responses. This valid and reliable scale demonstrated good internal consistency using confirmatory factor analysis and the Rasch model. Survey responses show that 73.3% of respondents utilize PROTS in practice and 26.7% refer patients out for PROTS. Survey responses show that, among US DCs, 91.9% indicate PROTS has value beyond identification of pathology, 86.7% indicate that PROTS is important regarding biomechanical analysis of the spine, 82.9% indicate that PROTS is vital to practice, 67.4% indicate that PROTS aids in measuring outcomes, 98.6% indicate the opinion that PROTS presents very low to no risk to patients, and 93.0% indicate that sharing clinical findings from PROTS studies with patients is beneficial to clinical outcomes. The results of the study indicated that based on clinical experience, the majority of DCs find PROTS to be vital to practice and valuable beyond the identification of red flags.

## 1. Introduction

Doctors of Chiropractic (DCs) are portal of entry healthcare providers trained in the diagnosis and management of spinal related conditions, with an emphasis on biomechanical dysfunction, in addition to screening for pathology. Interestingly, an estimated 85% of chronic low back pain cases are diagnosed as “non-specific low back pain”, not as a result of injury, but as a result of an unknown cause, typically from spinal biomechanical dysfunction [1]. DCs offer safe [2,3,4], non-pharmaceutical, non-surgical approaches to musculoskeletal conditions that have been shown to reduce opioid usage [5,6,7,8] and decrease surgical intervention [9,10] and disability [11,12,13] when compared to other therapies. Research shows that when a patient sees a DC first after a low back injury, surgical intervention is reduced to 1.5% compared to 42.7% when initially evaluated by a surgeon even after considering other important variables [9]. Additionally, chiropractic intervention has been shown to reduce opioid usage by 56% [8], and a survey taken in 2012 found “over 96% respondents with spine-related problems who reported the use of chiropractic manipulation stated that the therapy helped them with their condition” [14]. While there are many different methodologies utilized within chiropractic to determine care, plain radiography has a long history of utilization by the profession as a viable tool in assessing spinal dysfunction [15]. Current chiropractic scope of practice, while varied from state to state, allows DCs to order, perform, and interpret radiographs for various reasons including the evaluation of musculoskeletal disorders, red flags, biomechanical analysis and to aid in patient management [16].

Plain radiography of the spine (PROTS) is utilized in healthcare to help practitioners identify suspected red flags, is easily accessible, quick, and is valuable in the diagnosis of various conditions; however, there are limitations [17]. PROTS has limited ability to detect soft tissue injury, may lead to unnecessary procedures due to incidental findings and only produces single, flat images that lack detailed views of three-dimensional structures [18]. Despite recognized value, there is debate within the literature regarding red flags [19,20], pathology, safety [21], and spinal biomechanical analysis [22,23,24,25,26,27]. These debates have led to diametrically opposing viewpoints within the chiropractic profession on the risks, ethics and economics of ionizing radiation exposure, as well as the value PROTS provides in cases of mechanical spine pain. As a result, there are efforts to alter clinical guidelines regarding PROTS suggesting plain radiography be limited to only red flags (history of recent trauma, infection, cancer, failure to respond to treatment, neurological deficits, chronicity, etc.) [28], as well as guidelines suggesting PROTS should be expanded for the qualitative and quantitative assessment of the biomechanical components of the spine in addition to red flags [29]. Currently, individual interpretation and implementation of clinical research have created diversity in clinical opinions regarding PROTS, leading to different clinical experiences throughout the chiropractic profession, which to date have not been explored.

Clinical opinion and experience are important components of Evidenced-Based Practice (EBP). EBP was implemented with the goal of improving and evaluating patient care [30] and has rapidly gained acceptance in the developed world [31]. The EBP model requires the integration of three factors: robust research evidence, clinical expertise, and patient values [32,33]. A doctor’s clinical opinion, experience, and expertise, including the knowledge, judgment, and critical reasoning acquired through training and professional experience, are essential for implementing evidence-based practices [32,33]. Despite evidence supporting chiropractic care for mechanical spine pain, there is limited research regarding DCs’ clinical opinion and experience regarding the utilization of PROTS, which may be of value in the management of mechanical spine pain.

A 2020 report regarding US chiropractic practices reviewed six analyses conducted between 1993–2015, where 3810 US-based DCs were surveyed on whether the practitioner took radiographs in their office [34]. The report states that 47% of respondents took radiographs in their office and that 56.2% of their patients were radiographed. The 53% that did not take radiographs in their office referred 21.9% of their patients out for radiographs. Additionally, DCs that relied on radiographs were asked the frequency at which they performed or referred out for radiographs, as well as for repeat or follow-up radiographs to monitor patient progress or response to care (61.3%); to identify or rule out fracture, dislocation, and other pathology (94.1%); and to review for the possible presence of spinal displacements and (or) vertebral subluxation (66.8%). While this 2020 study had a large sample and included multiple questions regarding use of radiographs in chiropractic practice, the binary nature of the questions limited respondents’ ability to convey the extent of their preferences regarding utilization of plain radiography. A 2021 survey suggests that the DCs’ preference towards use of radiographs correlated with their view on DCs’ role in healthcare; however, the limited response options and few questions related to PROTS are insufficient to conclude the comprehensive opinions and experience of DCs use of PROTS [35]. Another study from 2017 surveyed a select group of 190 members from the American Chiropractic College of Radiology, known as chiropractic radiologists, and only had 73 respondents [36]. The study is limited by the small sample size of chiropractors and possible bias from members belonging to a select organization.

The current study was comprised of a combination of binary and Likert questions given to practicing DCs. The aim of the study was to examine the chiropractic profession’s opinions on the utilization of PROTS, based on their experience, in an attempt to provide the most in-depth evaluation of such opinions to date. As a result, the authors coordinated with a statistician in the development of the Clinician Opinion & Experience on Chiropractic Radiography (COECR) scale. Our hypothesis for the survey, due to recent national association press releases [28], is that the survey results would be heavily weighted against the utilization of PROTS and that we would see conflicting responses.

## 2. Materials and Methods

In creating the survey for this study, the authors intended to be consistent with EBP and developed a series of 11 questions to adequately reflect the clinical opinion and experience of the US DCs on the utilization of PROTS in a chiropractic clinical setting. All procedures performed in this study were in accordance with the ethical standards of the 1964 Helsinki declaration and its later amendments or comparable ethical standards. The authors developed 8 revisions of the survey over a period of 6 months with the goal to design neutral survey questions that represent the clinicians’ decision process and to minimize bias. The survey included a variety of both Likert and binary response options that could accurately reflect each DCs’ clinical opinions and experience rather than solely binary response options. Many aspects of the clinicians’ decision process were considered during the process, including why practitioners would or would not order PROTS, what value DCs attribute to PROTS, how to adequately reflect DCs’ clinical opinion and experience of PROTS as it relates to patient care, and issues related to plain radiography utilization safety and research.

The first question requested the participant’s name and address. Participants were notified in the introduction of the survey that their personal information would remain anonymous; identifying information was only available for the data analyst to ensure that survey responses were not duplicated since some DCs may have received multiple survey invitations. The intention to use the response for future publications was clearly indicated. If participants consented to participate, they proceeded to questions 2 through 11.

Questions 2 through 9 surveyed the participants about reasons that DCs may or may not order plain radiography and to determine the presence of radiographic equipment in the office. Additional findings from these questions include determining the level of agreement to utilization on a graded scale, questions pertaining to how, why, and when these procedures are valuable, how and if they have value as it relates to the direction and outcome of patient treatment and care, and to assess the clinical opinions and experience regarding the level of risk or safety associated with these procedures.

The 11th and last question of the survey was an open-response textbox that allowed DCs to explain the rationale that guided their respective answers to survey questions and to provide additional comments on the subject matter. A summary is provided in Appendix D, but these responses are not included in the analysis of this publication and will be reserved for future projects.

### 2.1. Survey Distribution

To gather a broad sample, we aimed to distribute the survey to as many US DCs as possible using three main data sources. The survey was distributed through email invitations (Appendix A) on multiple occasions during a period of 18 months between 2019 and 2021. The survey was formatted using SurveyMonkey, a cloud-based software platform developed to support online survey data collection. Although sampling weights were not developed, distribution of emails by state was provided from the database source.

The primary source of survey distribution was through an opt-in email database purchased from a leading chiropractic magazine publication resulting in email invitations on two separate occasions to the database, which included 49,747 DCs located throughout the US. Appendix B shows the distribution numbers and percentages based on the state of residence.

Next, the survey link was distributed utilizing an opt-in email source from a private company database. The survey link was distributed to approximately 20,400 US DCs on two separate occasions. Third, the survey was disseminated to the non-profit organization Chiro Congress (formerly known as the Congress of Chiropractic State Associations, COCSA) email database. Chiro Congress was selected since it is a national organization that has affiliated state chiropractic associations from nearly all US states. The state associations affiliated with the Chiro Congress network were invited to send the survey link to their members. Lastly, survey recipients were encouraged to share the survey with their colleagues: all survey email invitations stated, “Please send this survey link and encourage your colleagues to take the survey as well”.

Overall, the survey was emailed to an estimated 50,000 unique email addresses of licensed DCs. Approximately 5788 DCs opened the survey (an open rate of approximately 3.99–9.20%), of which 4301 DCs completed responses. If a DC submitted a survey response more than once, only the most recent submission was used in the analysis. Although the aim of the survey was to collect responses from practicing DCs in the US (*n* = 3641), some responses came from Canada (*n* = 459) and the rest of the world (*n* = 201). The survey submissions were widespread across the US, as represented in Appendix C.

### 2.2. Statistical Analyses

#### 2.2.1. Procedures and Sample

The analysis included all 10 quantitative survey questions (question 11 was qualitative and omitted from the analysis of the current study). Question 1 (Q1) collected demographic information; questions 2 (Q2) and 10 (Q10) had 5 response options each and allowed participants to choose multiple responses. Question 7 (Q7) offered 5 different categorical responses, limiting the participants to a single-category response. The remaining questions collected responses on a 5-point Likert-type scale.

To prepare the data for the analyses, variables were re-coded to account for item structure and to ensure that higher values corresponded to stronger clinical opinions. The dependent variable, Q7, originally contained 5 response options, but was dichotomously re-coded. Q2 and Q10 were dummy-coded, turning each response option into a stand-alone survey item. Dummy coding is a recategorization of discrete variables into a series of dichotomous items to ensure a linear relationship [37]. Additionally, by re-coding Q2 and Q10, the overlapping responses were eliminated. The remaining questions were re-coded in a way that higher values represented higher levels of the variable. Specifics about re-coding are detailed in the measures section.

#### 2.2.2. Dependent Variable

The utilization of plain radiography in chiropractic practices was assessed by the following survey item: Please select one answer that best describes your use of general spinal radiography in your practice. (This is NOT regarding advanced imaging such as CT/MRI). Respondents were provided with 5 response options to indicate use of spinal radiography:(1)I do NOT take radiographs in my clinic, I refer patients out to another facility (coded as 0);(2)I DO have an X-ray machine in my practice, but I still refer patients out to another facility for the majority of my spinal radiographs (coded as 0);(3)I have a plain film X-ray system in my practice and use it for the majority of my radiographs (coded as 1);(4)I have a DR (digital radiography) digital X-ray system in my practice and use it for the majority of my radiographs (coded as 1);(5)I have a CR (computed radiography) digital X-ray system in my practice and use it for the majority of my radiographs (coded as 1).

#### 2.2.3. Predictors

Appendix D provides the full list of survey questions that were used as predictors.

### 2.3. Clinician Opinion & Experience on Chiropractic Radiography (COECR) Scale

Q2.1–Q2.5 and Q10.1–Q10.5 (Table 1) establish a set of items that capture the clinical opinions of practicing DCs towards using spinal radiology in chiropractic practices. In an effort to construct a new scale, this study subjected the COECR scale to psychometric evaluation by investigating internal consistency of the scale, performing confirmatory factor analysis, and specifying and testing an Item-Response Theory (IRT) model [38,39,40].

### 2.4. Analytic Plan

The statistical analysis began with an examination of descriptive statistics and evaluating group differences using chi-square tests and *t*-tests. For categorical variables, percentages were reported. For continuous variables, the averages and standard deviations were described. Subsequently, binary logistic regression with logit link [41] was estimated and tested using the survey data. Binary logistic regression is a regression model where the outcome is following a binomial distribution with predictors of any form including continuous, categorical, or both [42]. Preference was given to binary logistic regression due to the distribution of the dependent variable (Q7). The descriptive statistics, chi-square tests, and logistic regression analyses were conducted in Statistical Package for Social Sciences (SPSS) version 24.0 [43].

To further understand DCs’ utilization of plain radiology, we performed Rasch analysis on the set of items that capture DCs’ clinician opinion and experience on PROTS. Fitting the Rasch model [44] to the data, we were able to evaluate the difficulty of endorsing each item included in the analysis as well as to estimate the relationship of each item with the underlying latent trait [45]. In this study, the latent trait, denoted by θ, is the clinical opinion towards DCs’ utilization of plain radiography in chiropractic practice.

### 2.5. Study 1: Predicting the Use of Plain Radiography of the Spine in Chiropractic Practice

A multiple binary logistic regression was estimated and tested to understand the relationship between DCs’ utilization of plain radiography of the spine in chiropractic practice and the rest of the variables included in the survey. Logistic regression can be considered an approach similar to multiple linear regression (with Gaussian outcome) except that the dependent variable is binary [46].

### 2.6. Study 2: COECR Scale Development and Validation

The Cronbach’s coefficient alpha [47] statistics was used to estimate the internal consistency of the COECR Scale. A one-factor CFA model was applied to responses Q2.1–Q-2.5 and Q10.1–Q10.5 in order to ensure the unidimensionality of the scale. Mplus 8.6 [48] was used to conduct the dimensionality study. To evaluate the statistical fit of the CFA models, we used likelihood ratio tests [49], chi-square tests, and model fit indices available for latent variable modeling. Following the recommendation of Hu and Bentler [50], we used the root-mean-square error of approximation [51], the comparative fit index [52], and normed fit index [53].

#### Rasch Model

The goal of this analysis was to evaluate the psychometric properties of the items on the COECR scale. Item Response Theory [54] was selected as the framework to test the psychometric qualities of items. Two important assumptions made for IRT models are unidimensionality [55] and local independence [56,57]. The assumption of unidimensionality was tested with CFA. To test the assumption of local independence, Q3 statistics were estimated [58]. The Q3 statistics are pairwise residual correlations after fitting the Rasch model for every item pair on a scale. Yen [59] provided the guidance for the assessment of local independence: “The expected value of Q3, when local independence holds, is approximately −l/(*n* − l)” (p. 198). The Q3 statistics for COECR ranged from −0.37 to 0.26, indicating that the assumption of local independence was not violated [59,60]. The Rasch model was estimated and graphed in R Statistical Software using irtoys and eRm [61]. The Q3 statistics were estimated using the mirt package [62].

## 3. Results

### 3.1. Initial Results

A summary of the overall 4301 survey respondents is provided (see Table 2 and Table 3). Not all respondents responded to every question. While the survey was intended to focus on US licensed DCs, there were additional respondents from Canada and other countries. The collected data allowed for statistical analysis of US responses, comparative analysis between US and non-US DC responses, and comparative analysis between those respondents with radiographic facilities and those who lack them as described in the methods section.

### 3.2. Descriptive Statistics, Chi-Square, and Mean Differences

A complete case analysis (also known as listwise deletion) for missing data was performed. Listwise deletion is a method that excludes an entire record from the analysis if any single value is missing [63]. The analysis revealed 232 cases with missing responses, which were removed from the dataset. Exclusion of cases with missing data on key variables resulted in an analytic sample of *n* = 4069. There were no systematic differences between the original and analysis samples.

Practicing DCs who obtain spinal radiographs using radiology devices/machines within their practice (*n* = 2985) compared with those who do not have radiology devices/machines in their practice who subsequently refer out to other facilities for PROTS (*n* = 1084) were systematically dissimilar on a number of predictors. Pearson chi-square tests revealed that a country where DCs practiced is a significant predictor of utilizing plain radiography. Additionally, utilizers versus non-utilizers of spinal radiology significantly differed on all but one COECR scale item (Q10.5; Investigate red flags [fracture, neurologic deficits, suspected pathology]). The results for these comparisons are presented in Table 4.

The factor analytic model showed that the items considered for the COECR scale assess a single trait—clinical opinion and experience regarding the utilization of PROTS in chiropractic practice. This is a prerequisite for establishing a unidimensional scale. The only item that impacted the factor in a different (negative) direction was “*I order radiography only for pathology and red flags*.” There is an inverse relationship for doctors that only order radiography for pathology and red flags and the total score on the scale. This means that DCs scoring higher on this item will score lower on the entire scale and vice versa. The results of this survey demonstrate that DCs who only order radiography for pathology and red flags responded in opposition to the other respondents to the survey (see Table 5).

Table 5 depicts the differences between utilizers and non-utilizers of plain radiography as a function of continuous variables included in the survey (Q3, Q4, Q5, Q6, Q8, and Q9).

Independent *t*-tests revealed that the DCs who were more likely to have X-ray units in their office believed that their educational and clinical experiences should allow them to decide whether to utilize plain radiology (*p* < 0.01; Q3); believed that patient outcomes would benefit from continued research regarding appropriate utilization of spinal radiology (Q5); were confident that DCs’ clinical experience together with patient preferences are adequate and appropriate for recommending PROTS (Q6); and/or believed that sharing spinal radiographic findings with the patient is beneficial for patient outcomes (Q9).

DCs who believed that PROTS may be risky to a patient’s health were less likely to have an X-ray unit in their chiropractic office (*p* < 0.01). Finally, DCs who believed that they should equally consider all three EBP components when making clinical decisions to obtain spinal radiographs were not statistically different from those who did not believe this in terms of having plain radiography in their office. The effect sizes for statistically significant results ranged from small to large (see Table 6).

### 3.3. Study 1: Logistic Regression

Binary logistic regression was estimated and tested to determine which of the variables remain predictive of DCs’ utilization of PROTS after controlling for the variability associated with all other predictors. Therefore, a binary logistic regression model with Q7 being the outcome variable and 18 categorical and continuous predictors was estimated and tested. For the country of practice, the US and Canada were included in the model while the rest of the world category (outside US and Canada) served as the reference group. For categorical variables, the lower level (coded as 0) served as a comparison level for the levels coded as 1. The continuous predictors were included in the model as continuous variables.

The overall model revealed statistical significance. The model was evaluated using Nagelkerke pseudo R^2^. All predictors, taken together, accounted for 38% of the variability in utilization of spinal radiography. The classification of cases was acceptable as the model correctly classified 48.6% and 94.5%, respectively, of DCs who utilized PROTS in their chiropractic practice and those who did not. The overall classification was 82% and was calculated using the following equation:$$P\left(\mathrm{correct}\;\mathrm{classification}\right)=P\left(y=1\;and\;\hat{y}=1\right)+P\left(y=0\;and\;\hat{y}=0\right)=P\left(\hat{y}=1|y=1\right)P\left(y=1\right)+P\left(\hat{y}=0|y=0\right)P\left(y=0\right)$$
which is a weighted average of sensitivity and specificity [64].

As presented in Table 7, DCs in the US are seven times more likely to utilize plain radiography, *OR* = 7.36, *p* < 0.01, compared to the rest of the world (outside US and Canada), while DCs practicing in Canada are three times more likely to utilize PROTS, *OR* = 3.17, *p* < 0.01. The DCs who utilized PROTS believed that:(1)radiographic procedures in a chiropractic office have value beyond the identification of pathology, *OR* = 1.54, *p* < 0.05(2)radiographs for biomechanical procedures have significant value, *OR* = 1.72, *p* < 0.01(3)radiographic procedures are vital to chiropractic care, *OR* = 5.93, *p* < 0.01(4)radiographic procedures aid in the measurement of clinical outcomes, *OR* = 2.23, *p* < 0.01(5)that sharing chiropractic clinical findings from radiographic studies with the patient is beneficial to their clinical outcome, *OR* = 1.18, *p* < 0.05(6)biomechanical analysis or measurements of spinal alignment are valid reasons for obtaining spinal radiograph, *OR* = 1.4, *p* < 0.05(7)and care plan modification consideration is a valid reason to obtain a spinal radiograph, *OR* = 1.57, *p* < 0.01.

Respondents who believed that PROTS should be utilized only for pathology or red flags were much less likely to use PROTS for the chiropractic practice, *OR* = 0.38, *p* < 0.01.

### 3.4. Study 2: Scale Construction

CFA was conducted using CFA procedures for binary or categorical items. The one-factor model regressed categorical indicators on a single factor: clinician opinions and experience on chiropractic radiography. Despite the significant value of the chi-square statistics, the model produced a good fit to the data: RMSEA = 0.05, 90% CI = (0.04, 0.06); CFI = 0.97; TLI = 0.97. In general, the chi-square is not considered to be a practical fit index, because it is strongly affected by sample size [65,66]. The item loadings ranged from 0.71 to 0.97 and were statistically significant (*p* < 0.001). Initially, item Q2.3 revealed negative loading on the factor, which was consistent with the previous analyses. The item was reverse coded to ensure positive loading. The CFA results are presented in Table 7.

The internal consistency reliabilities were estimated using alpha coefficient [47]. The estimate with original coding of Q2.3 was α = 0.8. There was an increase in internal consistency after recoding Q2.3: α = 0.84. Both coefficients are high and in support of a unidimensional scale. Two Rasch models were specified and tested using R Statistical Software. The first model (M1) was estimated with Q2.3 being originally coded, while the second model (M2) was estimated with Q2.3 being reverse coded. The results for Rasch models are presented in Table 6. Two types of fit statistics—infit and outfit indices—were estimated to assess the fit of the derived scale to the data. The infit is sensitive to unexpected responses near the item, whereas the outfit is sensitive to unexpected responses far from the item [67]. In M1, Q2.3 showed a misfit with *Outfit MSQ* = 12.6. In M2, the infit/outfit mean square estimates for all COECR items were within their reasonable bounds; thus, the sequence is considered stable and scalable. Figure 1 and Figure 2 show the item characteristic curves (ICC) for M1 and M2. The ICC is an S-shaped curve that portrays the probability of endorsing an item as a function of the latent trait. For M2, the location characteristic (item difficulty level) was higher for Q2.5 (*b* = −0.84) and Q10.1 (*b* = −1.0), while the items that were easiest to endorse were Q10.5 (*b* = −3.23) and Q10.4 (*b* = −3.24).

### 3.5. Item Response Theory

The authors used the Rasch IRT model to rank the scale items by difficulty, a common step in scale development. Table 8 presents the results of the Rasch model fit and Figure 1 and Figure 2 present the graphical ranking of the items before and after the “*I order radiography for pathology and red flags*” is recorded. Once again, the Rasch model showed that DCs who utilize plain radiography of the spine *only* to rule out pathology in the presence of red flags are statistical outliers and are significantly inconsistent with the clinical opinion of the chiropractic profession.

### 3.6. Results Summary

There is an inverse relationship in the responses between DCs that do and do not utilize PROTS in their practice; however, the clinical opinion of US DCs who utilized PROTS in their practice assigns a high value to the utilization of PROTS (Table 5 and Table 6). These doctors believe PROTS to be safe and that the DCs’ clinical experience is adequate for recommending PROTS. (Q6) In summary, 77.6% of these US-based DCs indicated that PROTS has value beyond the identification of pathology, 79.1% indicate that PROTS is important regarding biomechanical analysis of the spine, 83.0% noted PROTS to be vital to chiropractic practice and 84.7% believe PROTS aids in measuring outcomes (Table 4).

## 4. Discussion

While there is not consensus on the use of PROTS for chiropractic patient management, we surveyed a national sample of DCs to help clarify this topic with the most extensive investigation into the clinical expertise of the chiropractic profession regarding utilization of plain radiography to date. We demonstrate that DCs embrace a spectrum of opinions, ranging from possible perspectives that, due to safety concerns, utilization of PROTS be limited to diagnosing pathology in the presence of red flags [68,69,70,71] to an absolute necessity of X-ray images to determine spine and biomechanical parameters [71,72,73].

EBP includes three categories: published literature, patient preference, and clinician experience. The clinical experience and expertise of the practicing DCs has been neglected in the development of guidelines using EBP guidelines despite being an essential component of the EBP. The results of this study demonstrate that DCs’ clinical experience and expertise regards PROTS as vital to practice, valuable for patient diagnosis, care, management, biomechanical assessment, and overall outcomes.

Recent evidence suggests that 73.4% of practicing DCs read peer-reviewed research about patient management between several times a day to about once a month, and an additional 22.8% of DCs review published research 1–6 times per year. Further, 40.2% of practicing DCs review best practice guidelines from several times a day to about once per month, with another 37.6% reviewing EBP about 1–6 times per year [34] (pp. 106–108). Given the increasing evidence that chiropractic care has a positive impact on spinal health, the clinical experience and opinion of DCs should be considered in any practice guideline development that utilizes EBP as a foundation. Recall that EBP is the equivalent balance of three components: (1) the best research evidence, (2) clinical experience, and (3) the patient’s preferences [74]. The best research evidence is clinically relevant, peer-reviewed research that has been conducted using sound methodology. Clinical experience refers to the clinician’s cumulative experience, education, and clinical skills in managing patient care, and patient preference refers to the patient’s unique concerns, expectations, and values. These three components should be taken into consideration in the decision-making process for patient care [75].

The results of the survey show that a majority of the respondents utilize plain radiography (73.3%) within their practice facility, with 82.9% believing that plain radiography procedures are vital for chiropractic care. Additionally, 17.1% of DCs who do not utilize PROTS in their clinics believed that plain radiography is vital for chiropractic care provided. These findings indicate that the majority of DCs consider plain radiography vital to practice (Table 4). Respondents were disaggregated into two groups, those who owned an X-ray machine in their clinic and those who did not, which served as the dependent variable. The authors of the study analyzed the responses of the two groups on every item in the survey (the analysis of dichotomous items is presented in Figure 1, and the analysis of continuous items is presented in Figure 2). The two groups were statistically dissimilar on almost every survey item, which may suggest that the decision of owning a radiograph system in a chiropractic clinic goes above and beyond affordability. Believing that utilization of PROTS is vital to the chiropractic case and these radiographic procedures aid in the measurement of clinical outcomes emerged as the strongest predictors of having a radiographic system in a chiropractic clinic. These analyses (see Table 4) confirmed and strengthened previous findings (see Table 2 and Table 3). In addition to analyzing the survey data and reporting the results, the authors of the study utilized the collected survey responses to construct a scale (a measurement instrument) for future use. The authors provided a valid and reliable scale with good internal consistency to assess clinical opinions toward the use of plain radiography in chiropractic clinical management and is so named the Clinician Opinion and Experience on Chiropractic Radiography (COECR) scale.

Results from the present study unveiled intriguing differences between those who choose to perform in-office PROTS and those who do not. The binary logistic regression analysis revealed that, aside from geography, the strongest predictor of having a radiography system in a chiropractic practice was an opinion that radiographic procedures are essential to chiropractic care: DCs who endorsed this item were 6 times more likely (*OR* = 5.9) to have a radiography system in their clinics. The second strongest predictor was the utilization of PROTS to aid measurement of clinical outcomes. DCs adhering to these views were twice as likely (*OR* = 2.2) to own a radiographic system. On the other hand, DCs who took radiographs only when suspecting pathology in the presence of red flags were much less likely to own a radiographic system in their clinic (*OR* = 0.4).

The data do not support the assumption that DCs who fully adhere to EBP are less likely to utilize PROTS for chiropractic case management. The strength of adherence to EBP was assessed by asking respondents if the clinical decision to obtain spinal radiographs should be based on all three EBP components. While the majority of respondents agreed or strongly agreed that all three components should be equally considered (77.7%), there were no statistically significant differences in utilization of plain film radiography between those who agreed that all three EBP components should be equally considered and those who did not (22.3%). Moreover, this question did not emerge as a significant predictor of the utilization of plain film radiography. These findings suggest that most DCs believe that all three components of EBP should be considered together.

Regarding the overall clinical opinions and experience of practicing DCs towards PROTS, 9.5% own a CR digital X-ray system in their practice; 47.7% have a DR digital X-ray system in the practice; 16.0% own a plain film radiography system; 2.1% do have a plain radiography system in their facility, but they refer the majority of patients out for PROTS; and 24.7% do not own any plain radiographic equipment in their chiropractic practice. It is evident from the survey that DCs who undervalue the use of PROTS are less likely to own a plain radiography system, while the reverse is true for those who value PROTS as they are likely to own a radiography system.

There is evidence in the literature to support the rationale for the high clinical opinions reflected in the survey regarding value of PROTS, the relevance to biomechanical analysis and the relationship to measuring the outcome. Radiographic measurements such as frontal and sagittal spine alignment are well demonstrated to be important factors in predicting spinal health, quality of life and neurological dysfunction [76,77,78,79,80,81,82]. Frontal and sagittal spinal alignment are also correlated with many other radiographic parameters, including thoracic and pelvic morphology [83,84,85,86] Altered spinal balance remains associated with higher mechanical load and dysfunctional movement patterns and is a possible source of increased risk of pain and degeneration [87].

It is important to note that the literature also suggests that many of these radiographic parameters such as sagittal cervical spinal alignment and posture can be corrected with conservative care and these corrections can be corelated with improved function and health outcomes [88,89,90,91,92,93,94,95,96,97,98,99,100,101,102]. Other studies indicate that conservative care can result in radiographic changes to sagittal lumbar spinal alignment and posture, which is correlated with improved pain scores and health-related quality of life (HRQOL) [103,104,105,106,107]. Research has demonstrated that abnormal sagittal thoracic spinal alignment can be corrected, which is correlated with improvement in the risk of falls, headaches, forced expiratory volume, and HRQOL [108,109,110,111,112,113,114,115]. Additional studies have demonstrated that conservative correction of cervical lordosis and forward head posture can be associated with increased HRQOL, reduced back pain, and improved nervous system adaptability [100]. Coincidently, similar studies exist within the orthopedic research suggesting that the utilization of PROTS to measure surgical correction of biomechanical parameters such as sagittal vertical axis, lumbar lordosis, sacral slope, pelvic tilt and pelvic incidence angle have a direct impact on predictive surgical outcome, improved Oswestry Disability Index and improved patient HRQOL [116,117,118,119,120]. The current research suggesting the ability to correct radiographic parameters provides rationale for the clinical opinion of DCs indicated in the survey that more research should be dedicated to radiographic utilization in practice.

### 4.1. Other Findings

Additional findings also offer important insights. Item Q2.3 (*I only order radiographs for pathology or red flags*) showed negative factor loading when a one-factor CFA model was considered. This shows that the item measures the opposite pole of the intended construct, suggesting a negative linear correlation between the observed item and the latent construct measured [121]. The issue was echoed when the initial Rasch model was considered in the estimate of difficulty (the item was the most difficult to endorse) and the values of MSQ. After the item was reverse coded, the fit of the CFA and Rasch models was improved.

When considering the COECR scale, all items are consistent in producing a total score except Q2.3 (prior to recoding), which may suggest a different mindset for practitioners who take radiographs only when pathology is suspected. Although a minority, they strongly believe that the prudent use of plain radiography does not significantly improve long-term management of chiropractic patients and that there should be limitations on utilization due to concerns about safety, ethics, and economics. According to our findings, this is not the view of today’s practicing DCs.

### 4.2. Strengths

Methodologically, this is the strongest study in the literature regarding clinical experience for plain radiography utilization. So far, no study reported in the last 10 years implemented this level of sophisticated statistical methods combining descriptive analyses, group differences, predictive modeling, factor analysis, and IRT. The only study that used predictive methods was that of Pearce et al. [122], but the model in the study was challenged by the limitations described in the previous sections. For the first time, researchers collected a sizable, national sample surveying clinical expertise of the chiropractic profession toward plain radiograph utilization for chiropractic case management. Although the data from previous studies show systematic dissimilarities in clinical opinions toward the use of PROTS for chiropractic management between DCs who own radiographs in their clinics and those who do not, the items were considered one by one. The predictive model developed by the researchers in this study considered all items at once. This approach allows for predictors to reveal statistical significance while controlling for all other variables in the model. Additionally, our study shows that those who agreed with the statement “*I order radiography for pathology and red flags*” are in the minority, which contradicts current trends in evidence-based practice recommendations. These guidelines tend to dictate the political stance for the rest of the profession using research that is highly susceptible to bias and that does not consider biomechanical analysis, treatment strategies, or patient outcome. This requires further research and visibility in the literature to improve the professional understanding of the value of plain radiography on patient outcome that is evident in the clinical opinion of practicing DCs.

### 4.3. Limitations

This study is an examination of DCs’ clinical opinions and experience on using plain radiography for chiropractic case management. The findings in this study are novel and important; however, limitations should be considered. Methodologically, our study’s design involved a non-experimental approach evaluating cross-sectional variables. Thus, causal relationships cannot be established between the predictors and the outcomes. As with many self-report surveys, there were limitations regarding sampling and response biases. While we informed participants that the survey was anonymous, results may have been affected by social desirability bias [123]. The authors did not seek outside expertise in the creation of the survey and did not utilize an initial pilot study; some respondents may have misinterpreted the definition of red flags which could have skewed some of the responses. Additionally, it is possible that some DCs have no interest in the topic of utilization of PROTS, which would have resulted in self-selection bias and a lack of representation from DCs who are not strongly opinionated toward PROTS. When considering the EBP definition of clinical experience [75], our survey lacked the ability to determine the level of expertise and experience as it did not consider items such as years in practice, practice setting, levels of advanced education, hours of study per week and the financial implications of owning a radiographic unit. The distribution of the survey link to licensed DCs was based on email lists that were purchased from publication distribution lists and organizations that had access to significant national distribution of professionals. Therefore, we did not assign a priori probabilities to all population units to be selected in the sample. Although we attempted to minimize subjectivity, the inference of the findings to the target population may be susceptible to bias [124].

Question-related subjectivity and bias were minimized by constructing questions that were neutral, answer options that were not leading, survey results that were anonymous, and ensuring the anonymity of the organizing group. That said, there is no single correct way to structure a question or provide response options. Different respondents may have had different perceptions of the same question, which may influence survey responses and inference of the findings to the target population [124]. Additionally, the dependent variable in the predictive analysis (Q7) did not ask DCs directly whether they do or do not support taking spinal radiographs for chiropractic case management, so measuring clinical opinion was captured indirectly. Despite the limitations, this study provides novel information about DCs’ clinical opinions toward utilization of PROTS. Researching the opinions and experience of practicing DCs may clarify the utilization of plain radiography in chiropractic.

### 4.4. Future Research

Respondents indicated a desire for the chiropractic profession to align itself with the current trends in healthcare and refine our understanding of how to better utilize radiographic interpretation in the prediction and management of spinal health. There is a need for expanded research from the chiropractic profession to help determine the efficacy of the clinical opinions represented in this survey. Continued research may include additional surveys, qualitative studies, and observational studies. Additionally, longitudinal comparative studies are necessary to help understand the impact of spinal correction as measured with PROTS on patient QOL. Cooperation and joint research with the orthopedic profession may be beneficial, as there are already many orthopedic studies related to PROTS and the relationship between surgical biomechanical correction of the spine on QOL.

## 5. Conclusions

This survey provides the most extensive insight into the clinical opinion of the US chiropractic profession regarding PROTS and suggests that the majority of the DCs consider utilization of PROTS to have value beyond the identification of pathology, to be vital to chiropractic practice and essential to biomechanical analysis. The US DCs who utilized PROTS *only* to rule out pathology in the presence of red flags are, in fact, statistical outliers in this study and may represent a minority of US DCs. A majority of the DCs also consider the doctors’ clinical experience and expertise, coupled with patient preferences, to be appropriate for recommending PROTS. Most DCs in this survey found that sharing spinal radiographic findings with the patient is beneficial for patient outcomes. All participants in the survey believed that patient outcomes would benefit from continued research regarding appropriate utilization of PROTS. The results of this survey clearly indicate the value of PROTS reflected by DCs and demonstrate the need for continued research to help understand how this value can affect the quality of care, conservative correction of spinal alignment and patient health.

## Figures and Tables

**Figure 1 jcm-12-02169-f001:**
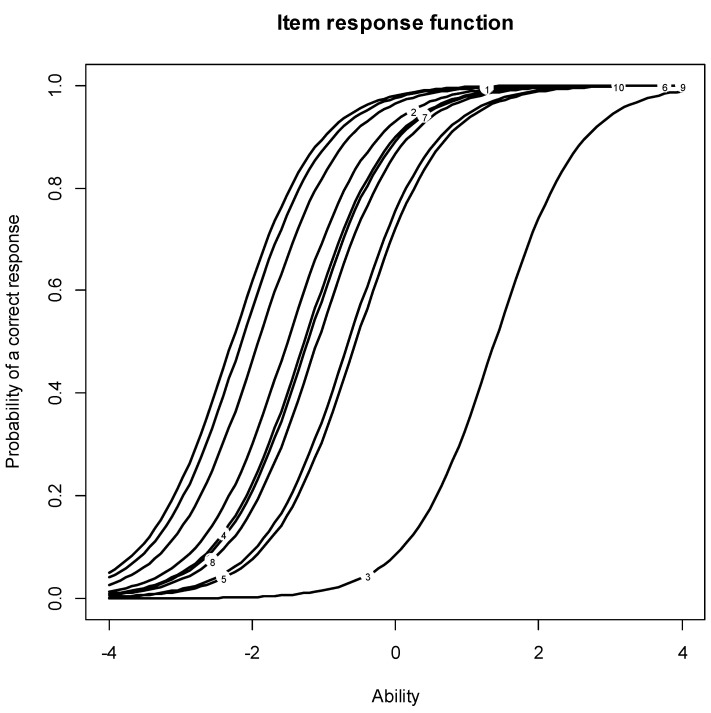
IRT Characteristic Curves for Survey Items.

**Figure 2 jcm-12-02169-f002:**
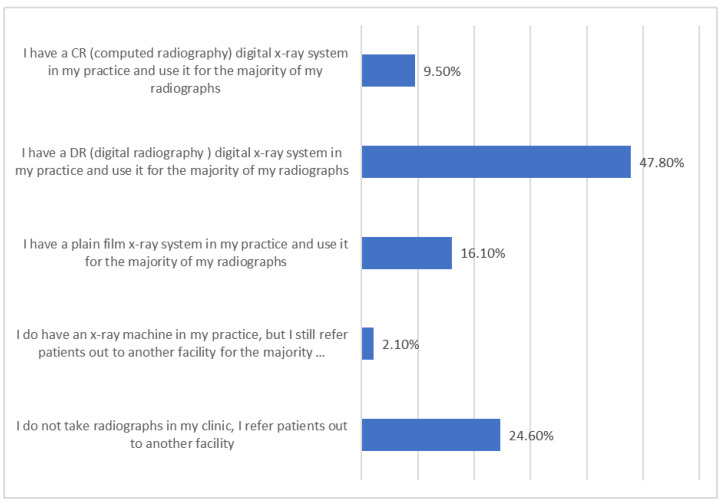
Original (Before Recoding) Distribution of Item Q7.

**Table 1 jcm-12-02169-t001:** List of survey questions and response options.

Q2.1: Radiographic procedures in a chiropractic office have value beyond identification of pathology.		0 = No	1 = Yes		
Q2.2: Radiographs for biomechanical analysis have significant value.		0 = No	1 = Yes		
Q2.3: I order radiographs only for red flags or pathology.		0 = No	1 = Yes		
Q2.4: Radiographic procedures are vital to the chiropractic care I provide in my clinic.		0 = No	1 = Yes		
Q2.5: I utilize radiographic procedures to aid in the measurement of clinical outcomes.		0 = No	1 = Yes		
Q3: What is your level of agreement/disagreement with the following statement: Based on the educational training and past clinical experiences, the Doctor of Chiropractic should be able to make their own clinical decision regarding the utilization of spinal radiographs on their patients?	1 = Strongly disagree	2 = Mostly disagree	3 = Neutral	4 = Mostly agree	5 = Strongly Agree
Q4: The foundation of an Evidence-Based Practice (EBP) is based on 3 integrated components: (1) Doctor’s Clinical Expertise, (2) Patient Preferences/Values, and (3) Best Research Evidence. When making the clinical decision to obtain spinal radiographs of your patient, should all three EBP components be equally considered?	1 = Strongly disagree	2 = Mostly disagree	3 = Neutral	4 = Mostly agree	5 = Strongly Agree
Q5: What is your level of agreement/disagreement with the following statement: In my clinical opinion, patient outcomes would benefit from continued research regarding appropriate utilization of spinal radiographs in the practice of chiropractic?	1 = Strongly disagree	2 = Mostly disagree	3 = Neutral	4 = Mostly agree	5 = Strongly Agree
Q6: What is your level of agreement/disagreement with the following statement: In the absence of published chiropractic research evidence, the doctor’s clinical experience combined with patient preferences are adequate for the appropriate recommendation of spinal radiographs in the practice of chiropractic?	1 = Strongly disagree	2 = Mostly disagree	3 = Neutral	4 = Mostly agree	5 = Strongly Agree
Q8: What level of risk do you believe is present in your chiropractic practice affecting your patients’ health, as a result of X-ray radiation from your utilization of radiography?	1 = No risk	2 = Very low risk	3 = Low risk	4 = Moderate Risk	5 = High risk
Q7.1: I do not take radiographs in my clinic. I refer out to another facility.					
Q9: What is your level of agreement/disagreement with the following statement: In my clinical experience, sharing chiropractic clinical findings from radiographic studies with the patient is beneficial to their clinical outcome?	1 = Strongly disagree	2 = Mostly disagree	3 = Neutral	4 = Mostly agree	5 = Strongly Agree
Q10.1: To determine adjusting technique or vertebral levels to be adjusted.		0 = No	1 = Yes		
Q10.2: Mechanical analysis or obtaining measurements of spinal alignment.		0 = No	1 = Yes		
Q10.3: Future plan modification and considerations.		0 = No	1 = Yes		
Q10.4: Determine spinal complications such as degenerative changes, anomalies, or defects.		0 = No	1 = Yes		
Q10.5: Investigate red flags (fracture, neurologic deficits, suspected pathology).		0 = No	1 = Yes		

**Table 2 jcm-12-02169-t002:** Summary of binary survey responses (Questions 2, 7, 10).

Predictor	%
Q2: Please select all statements that you agree with regarding spinal radiographs (multiple choices allowed). (*n* = 4231)	
Q2.1: Radiographic procedures in a chiropractic office have value beyond identification of pathology.	91.9
Q2.2: Radiographs for biomechanical analysis have significant value.	86.7
Q2.3: I order radiographs only for red flags or pathology.	16.5
Q2.4: Radiographic procedures are vital to the chiropractic care I provide in my clinic.	82.9
Q2.5: I utilize radiographic procedures to aid in the measurement of clinical outcomes.	67.4
Q7: Please select the one answer that best describes your use of general spinal radiography in your practice (This is not regarding advanced imaging such as CT/MRI). (*n* = 4138)	
Q7.1: I do not take radiographs in my clinic. I refer out to another facility.	24.7
Q7.2: I do have an X-ray machine in clinic, but I still refer patients out to another facility for the majority of my spinal radiographs.	2.1
Q7.3: I have a plain film X-ray system in my practice and us it for the majority of my radiographs.	16
Q7.4: I have a DR digital X-ray system in my practice and use it for the majority of my radiographs.	47.7
Q7.5: I have a CR digital X-ray system in my practice and use it for the majority of my radiographs. (CR digital requires the cassette to be placed into the image processor to process images)	9.5
Q10: Based on your clinical experience, which reasons are valid to obtain a spinal radiograph in the practice of chiropractic? (choose all that apply): (*n* = 4106)	
Q10.1: To determine adjusting technique or vertebral levels to be adjusted.	72.1
Q10.2: Mechanical analysis or obtaining measurements of spinal alignment.	84.5
Q10.3: Future plan modification and considerations.	81.9
Q10.4: Determine spinal complications such as degenerative changes, anomalies, or defects.	97.1
Q10.5: Investigate red flags (fracture, neurologic deficits, suspected pathology).	98.2

**Table 3 jcm-12-02169-t003:** Summary of responses to Likert-type scale items (Questions 3–6, 8, 9).

Predictor	*n*	Strongly Agree	Mostly Agree	Neutral	Mostly Disagree	Strongly Disagree
Q3	4223	92.9	5.3	0.7	0.6	0.5
Q4	4198	43.3	34.4	9.2	7.6	4.6
Q5	4188	60.9	21.4	11.0	4.4	2.3
Q6	4156	56.6	27.5	7.1	5.7	3.1
Q8 (No Risk-High Risk)	4138	0.2	1.1	10.1	61.2	27.4
Q9	4111	79.1	13.9	3.8	2.2	0.9

*n* = number of respondents per question.

**Table 4 jcm-12-02169-t004:** Percentage of respondents who do and do not obtain radiographs in their office.

		No Radiograph		Radiograph		
Predictor		%	*n*	*%*	*n*	*χ²*
Country						106.34 **
US		24.4%	842	75.6%	2603	
Canada		30.8%	137	69.2%	308	
Outside US and Canada		58.7%	105	41.3%	74	
Q 2.1						442.97 **
	No	76.4%	246	23.6%	76	
	Yes	22.4%	838	77.6%	2909	
Q 2.2						450.15 **
	No	64.3%	343	35.4%	188	
	Yes	20.9%	741	79.1%	2797	
Q 2.3						603.52 **
	No	19.1%	649	80.9%	2751	
	Yes	65.0%	435	35.0%	234	
Q 2.4						950.1 **
	No	73.9%	510	26.1%	180	
	Yes	17.0%	574	83.0%	2805	
Q 2.5						564.44 **
	No	50.5%	663	49.5%	650	
	Yes	15.3%	421	84.7%	2335	
Q 10.1						407.49 **
	No	49.1%	558	50.9%	578	
	Yes	17.9%	526	82.1%	2407	
Q 10.2						551.13 **
	No	64.5%	409	35.5%	225	
	Yes	19.7%	657	80.3%	2760	
Q 10.3						174.09 **
	No	46.0%	341	54.0%	400	
	Yes	22.3%	743	77.7%	2585	
Q 10.4						153.05 **
	No	75.0%	93	25.0%	31	
	Yes	25.1%	991	74.9%	2954	
Q 10.5						0.95
	No	31.3%	26	68.7%	57	
	Yes	26.5%	1058	73.5%	2928	

Note: ** *p* < 0.001.

**Table 5 jcm-12-02169-t005:** Average responses to clinical opinion questions.

Predictor	*n*	Mean	SD
Q3	4069	4.9	0.44
Q4	4069	4.07	1.12
Q5	4069	4.35	0.98
Q6	4069	4.3	1.02
Q8	4069	1.85	0.65
Q9	4069	4.68	0.73

*n* = Number of Survey Respondents. SD = Standard Deviation.

**Table 6 jcm-12-02169-t006:** Binary logistic regression model predicting position on radiographing chiropractic patients.

Predictor	*B*	*SE*	*Wald*	*OR*
Country				
US	1.99	0.17	130.88	7.36 **
Canada	1.16	0.2	33.18	3.17 **
Q 2.1	0.43	0.22	3.96	1.54 *
Q 2.2	0.54	0.18	9.15	1.72 **
Q 2.3	−0.98	0.13	58.52	0.38 **
Q 2.4	1.78	0.13	178.23	5.93 **
Q 2.5	0.80	0.11	57.18	2.23 **
Q 3	−0.02	0.11	0.03	0.98
Q 4	0.01	0.04	0.09	1.01
Q 5	−0.03	0.05	0.38	0.97
Q 6	0.06	0.05	1.82	1.06
Q 8	0.02	0.07	0.07	1.02
Q 9	0.17	0.07	5.27	1.18 *
Q 10.1	0.35	0.11	9.52	1.42 **
Q 10.2	0.34	0.16	4.48	1.4 *
Q 10.3	0.45	0.13	11.45	1.57 **
Q 10.4	0.41	0.31	1.74	1.51
Q 10.5	−0.32	0.31	1.04	0.73

*B* = Beta. *SE* = Standard Error. *Wald* = Wald test. *OR* = Odds Ratio. ** *p* < 0.001, * *p* < 0.05.

**Table 7 jcm-12-02169-t007:** Factorial structure of unidimensional model.

Item	Loadings	Standard Error
Q 2.1	0.95 **	0.01
Q 2.2	0.95 **	0.01
Q 2.3	0.71 **	0.02
Q 2.4	0.87 **	0.01
Q 2.5	0.82 **	0.01
Q 10.1	0.88 **	0.01
Q 10.2	0.95 **	0.01
Q 10.3	0.84 **	0.01
Q 10.4	0.97 **	0.01
Q 10.5	0.87 **	0.01

Note: *χ*² (35) = 1685.68, *p* < 0.001; RMSEA = 0.05, 90% CI (0.04, 0.06); CFI = 0.97; TLI = 0.97. ** *p* < 0.001. RMSEA = Root-Mean-Square Error of Approximation; CFI = Comparative Fit Index TLI = Tucker–Lewis Index.

**Table 8 jcm-12-02169-t008:** Item-level estimates and fit statistics for the Rasch model.

Item	*χ²*	*df*	*Difficulty*	*Outfit MSQ*	*Infit MSQ*
Q 2.1	1111.56	4110	−1.92	0.27	0.69
Q 2.2	1409.32	4110	−1.51	0.34	0.69
Q 2.3	51,793.5	4110	1.39	12.60	1.32
Q 2.4	2074.18	4110	−1.27	0.51	0.82
Q 2.5	2657.84	4110	−0.55	0.65	0.76
Q 10.1	2027.26	4110	−0.66	0.49	0.66
Q 10.2	1210.89	4110	−1.22	0.30	0.52
Q 10.3	1983.27	4110	−1.09	0.48	0.75
Q 10.4	773.515	4110	−2.15	0.19	0.65
Q 10.5	1707.58	4110	−2.28	0.42	0.86

*χ*^2^ = Chi-square. *df* = degrees of freedom. *Outfit MSQ* = Outlier-sensitive Fit Mean Square. *Infit MSQ* = Inlier-sensitive Fit Mean Square.

## Data Availability

All data is represented within the manuscript. Archived datasets analyzed during this study can be accessed publicly at the following link: https://radevidence.org/evidence-based-practice/ (accessed on 6 March 2023).

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
