# Peer review of "Examining Clinical Opinion and Experience Regarding Utilization of Plain Radiography of the Spine: Evidence from Surveying the Chiropractic Profession"

_jcm, 2023, doi:10.3390/jcm12062169_

Round 1

Reviewer 1 Report

I think this study has a serious flaw in the study design. The purpose of this study was to design a survey to assess chiropractic doctors’ opinions on utilization of plain radiography of the spine (PROTS). Based on this purpose, they constructed various survey questionnaires and validated them. Then, they concluded that “the results of this study demonstrate that “utilization of PROTS is safe and essential to chiropractic practice beyond the identification of pathology and red flags”.

First, I do not understand why designing this survey which asks chiropractic doctors’ opinions on PROTS is required to demonstrate that “utilization of PROTS is safe and essential to chiropractic practice”. The safety and necessity of PROTS can never be proved by this questionnaire-based survey. In addition, as a clinician, I do not think chiropractic doctors’ opinions on PROTS collected by a survey is clinically important or meaningful.

Second, the manuscript does not follow an appropriate format for an original study, and it was very hard to follow the flaw of the manuscript. The introduction section should be reduced to two or three paragraphs, removing unnecessary subheadings. The method and result section should also be reduced by removing unnecessary tables (Table 1 and 2; if necessary, please use appendix) and by simplifying tables (Table 3-7) using an abbreviation for each survey question. In addition, unnecessary explanations for several basic statistical methods, such as binary logistic regression analysis, Cronbach’s alpha, and Rasch model, should be removed. Statements regarding research ethics should be included in the method section.

Actually, the manuscript should be entirely re-written by referring to previous survey-based studies.

Reviewer 2 Report

Overview

The authors present a large survey examining mostly North American chiropractors’ beliefs regarding the use of radiographs of the spine which I enjoyed reading. The survey results found show that chiropractors frequently perceived radiographs to be helpful for purposes of biomechanical and structural clinical goals, and while chiropractors also used radiographs to detect pathology or red flags, this was seldom the only reason for obtaining radiographs. The authors’ survey also showed positive, significant correlations between markers of use of radiographs and ownership of radiographic equipment, which is an important and novel finding. Strengths of the survey include a large sample size, reliability calculations, and regression modeling. The survey provides interesting data and there has not been a survey like this which is as in-depth so I congratulate the authors for that. However, I think the authors should re-word several sections, add detail about clinical practice guidelines, and add more balance about potential negative aspects of overuse of radiographs so readers can understand all perspectives on the topic and see how their survey findings compare. I think if the authors improve the clarity of their findings and respond to these comments as expanded upon below, it will be a well-received publication in the chiropractic profession and beyond.

Major comments

·         The abstract is too long (it’s about 450 words I believe). According to MDPI’s author guide, to my understanding it should be about 200 words maximum.

·         Throughout the manuscript you reference “experience” and “expertise” including the title. However, you did not examine chiropractors’ experience or expertise per se, such as their years in practice, level of education, additional certifications, degrees, or specializations (the authors themselves have various letters/titles beyond a DC), or year of graduation, hours spent working per week, whether they are full-time or part-time, or work in an academic/teaching role, or what practice setting they work in (solo, multi-disciplinary, etc). Unless I’m missing something, I think you should delete “experience” and “expertise” from the title and manuscript. Really you are looking at chiropractors’ beliefs/opinion regarding PROTS regardless of expertise.

·         A lot of the manuscript focuses on the question about red flags even though this was a single question. I think the question was ambiguous as there are no examples of pathology or red flags. A recent systematic review from Yates shows that “red flags” can mean just about anything including a lack of response to care, age over 50, etc (https://link.springer.com/article/10.1007/s00586-019-06269-7). In addition, with a binary response it forces respondents to choose either Yes or No. Finally, the question is not valid as it uses the phrase “only” which is an absolute and survey psychology research states that absolutes should not be used in questions like this. I think you should reconsider your interpretation of the finding that so few people chose this response. Based on my example, some respondents may have chosen “yes” to only ordering radiographs for red flags if they considered the broad indications for red flags reported by Yates, whereas others may have only thought about cancer, infection, or fracture (a more limited view). Without defining red flags/pathology, you leave respondents to guess what that means. Please list this as a limitation as well as change some of the language regarding or simply say that few chiropractors reported use of radiographs only for the purpose of only evaluating for pathology or red flags. Throughout the manuscript you use the term “statistical outlier” to describe chiropractors who only order radiographs for pathology or red flags. Consider that this could be an error due to an invalid question and change the wording as above to simply say few chiropractors reported doing this. Why not instead re-iterate the other main findings of the study that chiropractors generally favored routine use of radiographs for biomechanical and structural goals? You can still mention red flags are seldom the sole reason for radiographs, but it seems odd to emphasize so strongly on that point.

·         Introduction – You need a paragraph explaining the current clinical practice guidelines for chiropractic. As there may be more than one set of clinical practice guidelines, maybe summarize the highlights as they pertain to your study. For example, what are the indications for PROTS per clinical practice guidelines? What about red flags? Biomechanical reasons? This will help frame your responses much better, as currently there is no description of this information.

·         Introduction – Also please include some of the suggested benefits and harms of PROTS to better frame your findings. Readers need context to understand why radiographs may be perceived to be beneficial or harmful in chiropractic practice. For example, they are used to assess structural changes, but there is a potential concern about over-identification of incidental findings, exposure to ionizing radiation, or increased cost, etc. Currently your manuscript seems more in favor of radiographs but for a scientific paper readers need to see a more even balance and you should remain objective to the literature. If desired you could mention the Choosing Wisely Campaign, but that may be too much detail.

·         I was a little confused about the distinction between “respondents who do and do not obtain radiographs in their office” and “doctors who utilized PROTS in their practice” – are these the same things? (e.g., see Table 4) You can be a DC and use radiographs in an independent imaging center but not in your actual office/practice setting, so is that considered using PROTS in the practice but not in the office? Can you make sure the terminology in the results section is clear regarding “office” vs. “practice”?

·         “the clinical experience and opinion of chiropractic doctors should be integrated into any practice guideline development that utilizes EBP as a foundation” – I don’t think you are able to suggest that your survey findings should change clinical practice guidelines. This is a bit preliminary. For this to happen you would need to show that increased routine use of radiographs improves patient care, not just that the majority of chiropractors use radiographs on a regular basis. Please delete any comments that CPGs should be changed. As someone who has helped create clinical practice guidelines for low back pain, the process typically involves higher level evidence than surveys. This information in your study could support a prospective study wherein routine use of radiographs are compared with the clinical outcomes where radiographs are used sparingly. If the results begin to favor routine use, then CPGs could change. However, based on opinion alone it seems preliminary to change CPGs.

·         Limitations – Please add that you didn’t consider respondents’ age, years in practice, education beyond a DC degree, or hours spent reading the scientific literature analysis, and these variables may influence their opinions towards use of radiographs. In addition, please add a sentence or two that ownership of radiography equipment may give a financial incentive to order radiographs, and that it may be a high startup cost to purchase radiography equipment for new DCs in private practice (if true). Also, I’m not sure if you considered chiropractors working in integrative settings such as hospitals, who typically won’t own their own equipment per se but may refer to an affiliated radiology clinic. As someone that works in a hospital, I can tell you that many large healthcare organizations have their own guidelines and appropriateness criteria and this can affect radiography use. Maybe none of these findings would change your results but they remain potentially important unmeasured confounders that readers should be aware of. I think you should cite a recent article summarizing chiropractors’ use of magnetic resonance imaging as well. If chiropractors occasionally use MRI, maybe this would also influence their use of radiography because it is a better tool for detecting pathology. (i.e., instead of only using radiography for red flags, they would select “no” because they would prefer to use MRI). If it’s justified, providers can actually skip radiographs and go straight to MRI.

·         Discussion – Please include a paragraph highlighting the differences between chiropractors’ beliefs/opinions in the survey and clinical practice guidelines. Are chiropractors practice habits consistent with these guidelines or are there discrepancies?

·         Conclusion – The conclusion is a little long, please shorten to about 200-250 words. Also the final statement seems to go above and beyond what the survey found – “Based on the components of EBP, the results of this study demonstrate that utilization of PROTS is safe and essential to chiropractic practice beyond the identification of pathology and red flags.” – Your study did not measure safety with radiographs or clinical care outcomes or constitute clinical evidence, so please rephrase to something like this -  “Many US chiropractors believe use of PROTS is safe and essential to practice beyond the identification of red flags”. Also, the statement “statistical outliers and are nonsignificant to the clinical opinion of the chiropractic profession” seems a bit harsh given this question may have been not valid. Please reword.

Minor comments

·         In the introduction (line 77-78) and elsewhere, you draw a distinction between previous studies regarding chiropractic & radiography where the prior studies used binary questions – “the binary nature of the questions limited respondents’ ability to convey the extent of their preferences regarding utilization of plain radiography”. However, the first five questions in your study appear to be binary as well, including an important question about radiographs for red flags. You could have asked, for example, that chiropractors rate their likelihood of using radiographs for the sole purpose of detecting red flags. Therefore, I think you should better describe your study as mixture of binary and Likert questions and likewise acknowledge the limitation of using binary questions.

·         In the Introduction – most of the paragraph beginning with “In 2005, the Joanna Briggs Institute (JBI) developed…” should be deleted or significantly reduced in length as it seems tangential to the focus of the study. I appreciated the last sentence in that paragraph “There is limited…” but that could be moved elsewhere in the introduction.

·         Please delete these lines in the introduction or move them to the Discussion as they are summarizing the results – “The current study explored chiropractic doctors’ clinical opinions, experience, and expertise and EBP parameters regarding utilization of plain radiography1 (17,23–25). This study reports the total 4,301 respondents results and further analyzes survey responses of the 3,641 U.S….”

·         Introduction – the paragraph beginning with “A 2021 survey aimed to describe characteristics of US-based Chiropractic Doctors…” This references a study by Gliedt et al. You analyze the study methods in too much detail then do not summarize its findings. Maybe summarize the key point that the authors found chiropractors’ preference towards use of radiographs correlated with their view on chiropractors’ role in healthcare. I agree that the previous study did not examine so many variables as you did, so you can highlight the research gap, but it can be more concise.

·         Please list somewhere in the introduction a basic statement about chiropractors’ scope of practice and what chiropractors do, for example that chiropractors are health care providers, portal of entry, often see patients with spinal disorders, and can order radiographs in the US

·         The hypothesis is a bit vague – “The hypothesis of the survey was that there would be a diverse opinion regarding plain film radiography that would help better understand various clinical opinions” – I can see what you are saying but can you quantify this somehow or make it more definitive or specific? Also, the final “opinions” is somewhat redundant. If you are seeing a lot of opinions, it will help you understand opinions. Can “opinions” not be used twice? Also, the study describes “examining” something in the title and throughout, but if you only have an exploratory hypothesis, it should be “explore” rather than “examine” throughout the manuscript, so this could change a lot. I think you can keep the hypothesis exploratory, but then you would have to change examining to “exploring”. If it’s more specific, then you can keep “examining”

·         Methods – Delete the equation for Cronbach’s alpha as this can be looked up easily. Instead, simply state the minimum threshold value you used for determining reliability.

·         Methods – are the equations for the Rasch model necessary? You provide the references so I imagine readers can look this up. It is slightly distracting and makes the paper much longer.

·         Methods – Did you test the survey on a sample population / pilot the survey, and calculate validity and reliability scores before disseminating it to the broader population? Unless I missed, it, this is not described. Also if you did pilot the study, you should describe the Cronbach’s alpha from the pilot testing in the Results as well.

·         Methods – Did you determine face validity with field experts to try to eliminate leading or confusing questions? Did you go through any revisions of the survey or how many times was it revised altogether?

·         Methods – Did you provide respondents with a paragraph or information sheet describing the purpose of the study along with the questions? This could be provided as a supplemental file or appear in the appendix.

·         Results – Table 3 – The first column of responses is not labeled, should it be “yes”? Otherwise it’s not clear if it’s Yes or No.

·         Discussion - “Radiographic measurements are well demonstrated to be an important factor in predicting spinal health, longevity, and neurological dysfunction such as frontal and sagittal spinal alignment” – Longevity does not make sense here (like how long people live? The references did not support that), consider stating “have been demonstrated to be predictive of quality of life, spinal health, and neurological dysfunction” – the examples you give could be deleted or moved to the front of the sentence, as there is a misplaced modifier (frontal/sagittal alignment are not examples of neurological dysfunction, rather, they are measurements)

·         The section regarding future research was a little confusing. You present several questions rather than research designs. It might be easier to read this if it was written in paragraph form and you stated if you were hoping to conduct additional surveys, qualitative studies, observational studies, or a prospective/experimental study. I don’t think you need so many questions/study ideas as well and currently #3, #5 can be deleted or re-worded and combined as they are redundant with #2. I think with some of the Discussion points, you would need a long-term prospective cohort study to examine health-related QOL in patients receiving radiographs at the start of chiropractic care versus those not receiving initial radiographs. This will better tell you if chiropractors’ beliefs hold true for clinical outcomes.

·         “There is strong evidence in the literature to support the rationale” – The evidence provided is not is considered strong evidence, it consists of narrative review papers and observational studies. You would need a systematic review or randomized controlled trial to represent a strong level of evidence. Please change “strong” to “some”.

Comments for clarity

·         Throughout – consider changing “utilization” to “use” for simplicity. “Utilization” also sometimes implies that you are repurposing something and using it for something it’s not intended for.

·         Introduction – “33 Million” – don’t need to capitalize Million

·         Introduction – “Nearly 4,000 US-based chiropractic doctors (n=3,810)” – can replace “nearly 4,000 with the sample size as it’s redundant

·         Consider abbreviating Chiropractic Doctors to DCs (optional). You use this term 78 times in the manuscript so it may shorten it substantially. Since this study mostly concerns the US, where chiropractors have a doctoral degree, I think this would be OK. If it’s not possible then the change is not needed. I’m not sure about Canada or the other countries included, and if these countries also have exclusively Doctors of Chiropractic or if they have a master’s degree or something else. You could also shorten to simply “chiropractors”

·         “Many aspects of the clinician’s decision process” – should be “clinicians’ ”

·         Figure 2 – Could you please remove the shaded background, and make the text not be in capital letters? Also consider making the % values at the end of the bars in black rather than white, so it can be seen more clearly. Also please explain acronyms in caption (even if redundant with main text) so it can be interpreted in a stand-alone manner.

·         Table 6 – you need a line at the bottom of the table indicating what the * and ** indicate, I imagine this is a p-value less than .05 or less than .01 or another value but it’s not clear.

·         “the decision of owning a radiograph in a chiropractic clinic” – should this say radiography equipment or system instead? Same comment applies to “predictors of having a radiograph in a chiropractic clinic.”

·         “Obviously, this is not the view” – change to “According to our findings, this is not the view…” – it may not be obvious to the readers

·         This section coped here should be or shortened to be more concise, consider saying that your findings suggest many practicing chiropractors hold beliefs regarding radiography divergent from those espoused by clinical practice guidelines. Also, as noted above, this type of critique of guidelines would require a manuscript unto itself. You haven’t at this point even said what the guidelines say – “These guidelines tend to dictate the political stance for the rest of the profession using research that is highly susceptible to bias and that does not consider biomechanical analysis, treatment strategies, or patient outcome. This requires further research and visibility in the literature to improve the professions understanding of the value of plain radiography on patient outcome that is evident in the clinical opinion of practicing chiropractic doctors.”

·         Table 3 – “dualization” – is this a typo?

·         Table 3 – high risk to low risk is written at the top, why not put those headings into the question itself as these risk categories are only used in one question. It’s confusing reading the headings as they don’t apply to most of the other questions. Just say (high / moderate / low / very low / no risk) after Q8 text on the left column.

·         Table 4 – there seems to be an issue with vertical alignment or position of text in each row. Starting with the row with “Radiographic procedures” - the text prompts on the left do not line up with the No / Yes and percentages while the Chi-squared value on the right column does line up.

·         The keywords are not true MeSH terms. Please go here and find MeSH terms to help with PubMed indexing: https://meshb.nlm.nih.gov/search?searchMethod=FullWord&searchInField=allTerms&sort=&size=20&searchType=allWords&from=0&q=chiropractic

·         Appendix A – can you order the percentage comments from highest to lowest? Currently it starts with 5% then 1%; can you make it 41%, then 37% etc.?

·         “increased telomere length” – please delete as this is not really a clinical topic

·         “Additional findings also offer important insights. Item Q2.3” – the subsequent statement is missing the “only”

Round 2

Reviewer 2 Report

I congratulate the authors on making several revisions to produce a much stronger manuscript, and quickly as well. I am glad to see several added and/or deleted paragraphs and added Limitations. I only have a few minor comments.

Minor comments

1.       Abstract conclusion – The last 2 sentences could be improved as the term “essential” was not in your survey itself (rather “value” or other terms) and I think you should continue the trend of emphasizing the findings in terms of the respondents beliefs/opinions/experiences rather than saying that PROTS is essential. Consider stating something voicing the beliefs/considerations of the surveyed (or US) population – feel free to modify this suggestion slightly: “Our findings suggest that most surveyed chiropractors considered utilization of PROTS to have value beyond the evaluation for pathology and red flags in practice” – finally I think the last sentence in the abstract about evidence based practice is a little superfluous, and should be deleted, as this is not your main finding and may distract readers from the major finding above.

2.       Introduction – “Plain radiography of the spine (PROTS) is generally accepted for the identification of red flags” – Please delete the “generally accepted for the identification of red flags” – this is illogical as most red flags come from the history and exam, and it’s not a generally accepted tool for identifying serious pathology (compared to MRI which is more accurate). Even if it is, you are sort of jumping ahead to the results/Discussion. I like the introduction elsewhere where it’s more balanced showing both opinions.

3.       I still think you should remove “expertise” from the title and abstract as you did not establish the survey respondents as “experts”. I agree with your reasoning to keep “experience” throughout the manuscript. This is not a “deal-breaker” on the publication acceptance, but changing it may improve the post-publication acceptance of your findings. The term “expertise” seems to establish the study population as experts, whereas professors or academics (who consider themselves the experts) might have a problem with this, especially if they did not answer your survey.

Comments for clarity

1.       Introduction - “respondants” should be “respondents”

2.       Introduction – “see a conflicting responses” – delete the “a”

3.       Last – simply a helpful reminder. Make sure the study abstract and other details which you modified match within the MDPI submission platform.
